# Metabolic Responses of Lung Adenocarcinoma Cells to Survive under Stressful Conditions Associated with Tumor Microenvironment

**DOI:** 10.3390/metabo14020103

**Published:** 2024-02-02

**Authors:** Angeles Carlos-Reyes, Susana Romero-Garcia, Heriberto Prado-Garcia

**Affiliations:** 1Laboratorio de Onco-Inmunobiologia, Departamento de Enfermedades Crónico-Degenerativas, Instituto Nacional de Enfermedades Respiratorias Ismael Cosio Villegas, Mexico City 14080, Mexico; 2Facultad de Ciencias, Universidad Nacional Autónoma de México, Mexico City 04510, Mexico; sugar_cia@comunidad.unam.mx

**Keywords:** tumor metabolism, tumor microenvironment, specific growth rate, lactic acidosis, hypoxia, glucose deprivation

## Abstract

Solid tumors frequently present a heterogeneous tumor microenvironment. Because tumors have the potential to proliferate quickly, the consequence is a reduction in the nutrients, a reduction in the pH (<6.8), and a hypoxic environment. Although it is often assumed that tumor clones show a similar growth rate with little variations in nutrient consumption, the present study shows how growth-specific rate (µ), the specific rates of glucose, lactate, and glutamine consumption (qS), and the specific rates of lactate and glutamate production (qP) of 2D-cultured lung tumor cells are affected by changes in their environment. We determined in lung tumor cells (A427, A549, Calu-1, and SKMES-1) the above mentioned kinetic parameters during the exponential phase under different culture conditions, varying the predominant carbon source, pH, and oxygen tension. MCF-7 cells, a breast tumor cell line that can consume lactate, and non-transformed fibroblast cells (MRC-5) were included as controls. We also analyzed how cell-cycle progression and the amino acid transporter CD98 expression were affected. Our results show that: (1) In glucose presence, μ increased, but q_S Glucose_ and q_P Lactate_ decreased when tumor cells were cultured under acidosis as opposed to neutral conditions; (2) most lung cancer cell lines consumed lactate under normoxia or hypoxia; (3) although q_S Glutamine_ diminished under hypoxia or acidosis, it slightly increased in lactate presence, a finding that was associated with CD98 upregulation; and (4) under acidosis, G0/G1 arrest was induced in A427 cancer cells, although this phenomenon was significantly increased when glucose was changed by lactate as the predominant carbon-source. Hence, our results provide an understanding of metabolic responses that tumor cells develop to survive under stressful conditions, providing clues for developing promising opportunities to improve traditional cancer therapies.

## 1. Introduction

Sustained proliferative signaling, evasion of growth suppressors, and replicative immortality have traditionally been considered hallmarks of cancer [1,2]. In contrast, normal differentiated cells are exposed to control systems that prevent aberrant individual cell proliferation [3]. Rapid tumor proliferation has been associated with a robust glucose and glutamine metabolism. Otto Warburg showed that even under normoxic conditions, tumor cells highly consume glucose while producing high levels of lactic acid. Warburg postulated that this phenomenon was due to mitochondrial failure, a phenomenon termed aerobic glycolysis [4].

With disorganized tumor proliferation, oxygen partial tension diminishes inside the tumor, and acid production increases; both hypoxia and acidosis (this last defined as an extracellular pH < 6.8) become selection forces that govern tumor cell adaption [5,6,7]. Tumor cells cope with different gradients of oxygen tension, glucose, and glutamine. These factors and fluctuations of lactate and acidosis give rise to a fluctuating tumor microenvironment. Although changes in lactate or oxygen levels have been studied [8,9], these factors have been considered individually. How this changeable tumor microenvironment affects the kinetic parameters of tumor and non-transformed cells, such as the growth rates, the specific rates of glucose and glutamine consumption, and lactate and glutamate production, still needs to be understood.

Our understanding of metabolic pathways has been primarily based on studies of non-proliferating cells in differentiated tissues [3]. Here, we evaluated the growth rates, the specific rates of glucose and glutamine consumption, and lactate and glutamate production to determine if lung tumor cells proliferate under stressful culture conditions associated with the tumor microenvironment. We compared normoxia vs. hypoxia, neutral pH vs. acidosis, and glucose vs. lactate as the carbon source. We determined how the glucose consumption rate of tumor cells varies when the tumor microenvironment changes. This allowed us to determine how tumor cells adapt to survive these stressful conditions. In addition, because the accessory protein 4F2hc (CD98) is associated with amino acid transporters (ASCT2 and LAT1/2) that play a crucial role in cell growth and survival [10,11], we analyzed CD98 expression in lung cancer cell lines cultured in the various tested conditions.

Thus, to determine how fast tumor cells proliferate compared to non-transformed cells, this report evaluates how culture conditions associated with the tumor microenvironment affect the kinetic parameters of lung cancer cells and compares these parameters with those of non-transformed cells under culture conditions related to the tumor microenvironment.

## 2. Materials and Methods

### 2.1. Cell Lines

Four human lung carcinoma cell lines, the adenocarcinoma cell lines A549 and A427 and the squamous carcinoma cell lines Calu-1 and SKMES-1, were used in this study. Additionally, this study includes, as controls, the breast cancer cell line MCF-7 because it consumes lactate without glucose [12] and the MRC-5 fibroblasts as proliferative non-transformed cells. All cell lines were obtained from the American Type Culture Collection (ATCC) (Manassas, VA, USA).

### 2.2. Cell Culture

The tumor cell lines and fibroblasts were cultured in RPMI-1640 (Sigma-Aldrich, St. Louis, MO, USA) supplemented with 10% heat-inactivated fetal calf serum (FCS, Hyclone, Logan, UT, USA), 100 μg/mL of streptomycin, and 100 U/mL of penicillin at 37 °C with 21% oxygen and 5% CO_2_. The cell lines grew in monolayers and were harvested by trypsinization.

### 2.3. Tested Culture Conditions

Each carcinoma cell line and fibroblasts were grown under different culture conditions, modifying the carbon source, pH, and two oxygen tensions. RPMI-1640 glucose-free medium (Sigma-Aldrich, St. Louis, MO, USA) supplemented with 10% heat-inactivated FCS, 100 μg/mL of streptomycin, and 100 U/mL of penicillin was also supplemented with glucose (10 mM) or L-lactate (20 mM). The pH was adjusted to 7.2 or 6.2 using HCl (37% *v*/*v*) in both cases. The initial concentration of glutamine ranged from 0.8 mM to 1.4 mM, depending on the batch number of the medium. Normoxic cells were incubated in a humidified chamber at 37 °C with filtered atmospheric air (21% O_2_) and 5% CO_2_. Hypoxic cells were incubated in a humidified Billups–Rothenberg chamber (Del Mar, CA, USA) with 2% O_2_, 93% N_2_, and 5% of CO_2_ at 37 °C.

### 2.4. Growth Kinetics of Tumor Cells

Two 24-well plates were seeded equivalently. One plate was used for normoxic conditions, while the other was used for hypoxic conditions. Six wells of each plate were seeded with the four different media formulations. Cells were seeded at a density of 1 × 10^5^ cells/mL for A427, A549, Calu-1, SKMES-1, and MCF-7 cell lines, whereas MRC-5 cells were seeded at 5 × 10^4^ cells/mL. A sample of the culture media from each of the wells was stored at −20 °C at the beginning of each kinetic experiment, representing time zero.

Plates were incubated for 96 h. Every 8, 12, or 24 h, depending on the cell line, the supernatant from each well was removed. Immediately, the volume of media present in each well was measured to account for evaporation in the analysis of metabolites. Cell-free supernatants were stored at −20 °C for later analysis. The cell growth and viability were evaluated every 8, 12, or 24 h, depending on the cell line. The cells were counted, and cell viability was determined with trypan blue dye exclusion using a TC20 Automated Cell Counter (Bio-Rad Laboratories, Inc., Hercules, CA, USA). All cultures were repeated at least twice.

### 2.5. Determination of Metabolites

At time zero for the metabolites determination, an aliquot of the initial culture medium was stored at −20 °C. Then, every 8, 12, or 24 h, an aliquot of the supernatant was taken until 96 h had been completed. The total volume of each well was measured, and cell-free supernatants were stored at −20 °C for later analysis [12].

The amounts of glucose, l-lactate, l-glutamine, and glutamate were determined using a YSI 2900 biochemistry analyzer (Yellow Springs Instruments, Yellow Springs, OH, USA) and membranes containing immobilized specific enzymes (d-glucose oxidase, l-lactate oxidase, l-glutamine oxidase, and l-glutamic acid oxidase; YSI, Yellow Springs, OH, USA). For each metabolite, particular standards were prepared and used according to the manufacturer’s instructions. The volume of evaporation was used to correct the quantity of each metabolite.

### 2.6. Evaluation of Cell Morphology

A427 cells were cultured under the above conditions using plates for confocal microscopy and then stained as described in [13]. Briefly, after 72 h incubation, cells were stained with MitoTracker Green 250 nM (Thermo Fischer Scientific, Eugene, OR, USA) to visualize mitochondria distribution. Cells were incubated for 30′ and washed with PBS. CellMask Orange (Thermo Fischer Scientific, Eugene, OR, USA) was added and further incubated for 5 min. After washing, cells were incubated with Hoechst 33342 (Thermo Fischer Scientific, Eugene, OR, USA). Digital images were acquired on an FV-1000 laser scanning confocal microscope (Olympus, Tokyo, Japan). FIJI v 2.14.0 was used for cell analysis.

### 2.7. CD98 Surface Determination

The amino acid transporter CD98 expression was evaluated by flow cytometry. Briefly, cell lines were cultured in 24-well plates under the above-described conditions for 48 h. The initial cellular concentrations per well were adjusted to 1 × 10^5^ cells/mL for A549, A427, and SKMES cell lines and 1.5 × 10^5^ cells/mL for Calu-1. All cultures were repeated at least twice. After incubation, the cells were harvested by non-enzymatic treatment (EDTA-MOPS). The cells were then washed with PBS-supplemented albumin (1% *w*/*v*) and sodium azide (0.1% *w*/*v*). The Zombie NIR fixable viability kit (BioLegend, San Diego, CA, USA) was used to gate viable cells. For immunostaining, the cells were incubated with monoclonal antibodies for CD98 (clone 5E5, FITC) from eBioscience (San Diego, CA, USA) at room temperature for 30 min. After incubation, the cells were washed and fixed with paraformaldehyde (1% *w*/*v*) for further analysis on a FACS Canto II Flow Cytometer (Becton Dickinson, Lincoln Park, NJ, USA). At least 10,000 events were acquired from the region of viable cells. The median fluorescence intensity (MFI) of CD98-FITC was determined. The results were analyzed with FlowJo V10 Software (Beckton Dickinson, Ashland, OR, USA).

### 2.8. Cell Cycle Analysis

A BrdU Flow Kit (BD PharMingen, San Diego, CA, USA) was used to determine the cell cycle phases and to measure the incorporation of BrdU into the DNA of proliferating cells, according to the manufacturer’s protocol. Briefly, 1 × 10^5^ A427 cells were seeded per well using 24-well tissue culture plates and cultured under the above conditions for 48 h. This was followed by adding 10 μM BrdU and an additional one-hour incubation. The cells were harvested by trypsinization, fixed using BD Cytofix/Cytoperm buffer, and permeabilized using Cytoperm buffer. The cells were then incubated for one hour with DNase at 37 °C. A 1:50 dilution of the fluorescein isothiocyanate (FITC)-conjugated anti-BrdU antibody in wash buffer was prepared and added to the fixed cells. Cells were incubated for 20 min at 37 °C, washed using BD wash buffer, and total DNA was stained with 7-amino-actinomycin D (7-AAD), followed by flow cytometric analysis. Culturing and staining were independently performed three times. Doublets and debris were excluded. Then, BrdU content (FITC signal) and total DNA content (7-AAD signal) were determined using FlowJo V10 software.

### 2.9. Statistical Analysis

All kinetic parameters were calculated during the exponential growth phase. At least two independent experiments were performed. All kinetic parameters were calculated as described in [12].

All values are reported as the mean ± s.e.m. Two-way ANOVA tests and analysis of variance were used where appropriate, with Bonferroni-corrected post hoc testing for a significance level of 0.05. Statistical analyses were performed using Graph Pad Prism software version 5.0 (La Jolla, CA, USA).

## 3. Results

### 3.1. Most Tumor Cells Increased Their Proliferation Rate under Acidosis when Glucose Was Available

We carried out the metabolic characterization of four lung tumor cell lines (A427, A549, Calu-1, and SK-MES-1), fibroblast cells (MRC-5), and a breast cancer cell line (MCF-7) in different culture conditions that emulate various tumor microenvironment conditions. To do this, we varied the carbon source, using either glucose (10 mM) or lactate (20 mM). A 20 mM lactate concentration was selected because this concentration is an intermediate concentration between the low and high limits found in some tumors [14].

We also varied the pH, using a pH of 7.2 or 6.2, because it has been reported that solid tumors frequently have an acidosis microenvironment [7]. Furthermore, we included normoxia and hypoxia conditions because it has been shown that hypoxia might support aggressiveness, metastasis, and failure of cancer therapy [15]. Table 1, Table 2, Table 3 and Table 4 summarize the kinetic parameters calculated during the exponential growth phase for each cell line and each condition. From the start of the experiment (time zero) to the beginning of the growth phase, there was an adaptation phase (lag phase) variable for each cell line and culture condition.

The adaptation phase of non-transformed MRC-5 fibroblasts lasted 24 to 48 h. Depending on the growth condition, the viable cells decreased by 30% to 50% during this period. Subsequently, the living cells recovered and began their exponential growth (Table 1 and Table 2). When MRC-5 cells were grown in glucose, the growth-specific rate (μ) diminished by 60% to 80% with acidosis conditions. When MRC-5 cells were grown using lactate at a pH of 7.2 and under hypoxia conditions (Lac/pH7.2/H), they could not proliferate, and their viability diminished. The μ was 3-fold higher with normoxic than with hypoxic conditions (Table 1 and Table 2).

When the A427 cell line was cultured using glucose, the cells did not undergo an adaptation phase; instead, they immediately entered the exponential growth phase. The µ was 80% higher in normoxic conditions compared to hypoxic conditions. Additionally, the µ was 60% to 70% higher in acidic compared to neutral conditions when cells were grown under normoxia or hypoxia, respectively (Table 1 and Table 2). Figure 1 shows changes in the morphology of these cells when cultured under different conditions. We previously showed that mitochondria content varies when changing the carbon source [13], so we included mitochondrial staining for the conditions tested. Cells cultured were smaller and rounded when cultured under pH 6.2, while mitochondria were aggregated. Under hypoxia, cells were more elongated while the mitochondrial content decreased. When cells were incubated with lactate and normoxia, cells were smaller and elongated, and mitochondrial content increased under acidosis (pH 6.2). When A427 cells were grown under Lac/pH 7.2/H or Lac/pH 6.2/H conditions, the cells did not proliferate but remained viable over 48 h.

The adaptation phase of A549 cells lasted for six hours. The growth-specific rate (μ) was higher in normoxic compared to hypoxic conditions. Additionally, the µ was 50% higher in acidic conditions than in neutral conditions when cells were grown using lactate as the predominant carbon source.

Calu-1 cells exhibited a different behavior than the other cell lines in this study. When Calu-1 cells were incubated with lactate-containing media, the cells remained alive without proliferating for over 30 h and subsequently began to die (Table 2 and Figure 2a,b). Thus, the growth of this cell line was significantly affected by the absence of glucose (Figure 2A,B,a,b).

The SKMES-1 cell line did not exhibit an adaptation phase. It was the only cell line in our study that could proliferate in all conditions tested (Figure 3A,B,a,b).

Likewise, MCF-7 cells did not exhibit an adaptation phase. The µ was higher in normoxic compared to hypoxic conditions. Interestingly, the MCF-7 µ value with lactate present was similar to the µ value with glucose under normoxia (Table 1 and Table 3).

In general, the µ of tumor cells was up to 15-fold higher than that of fibroblasts, depending on the culturing conditions. For most cell lines, the µ was 5% to 300% higher in normoxic conditions compared to hypoxic conditions. When cells were grown under hypoxia, cell proliferation in the presence of glucose was 20% to 90% higher with acidosis; in the case of lactate, acidosis allowed for survival or slight proliferation (Table 1, Table 2, Table 3 and Table 4).

### 3.2. Specific Rates of Glucose Consumption and Lactate Production of Tumor Cells Diminished under Acidosis

In all lung tumor cells, the specific rate of glucose consumption (qS Glucose) was 20% to 100% higher under hypoxic conditions compared to normoxic conditions. Additionally, the qS Glucose was up to 3-fold higher under neutral compared to acidic conditions, suggesting that acidosis regulates glucose consumption negatively. Surprisingly, MRC-5 cells cultured in glucose-containing media had the highest qS Glucose in comparison with the other tumor cell lines in this study (Table 3 and Table 4); accordingly, 1 × 10^6^ proliferating MRC-5 cells were able to consume glucose faster in comparison to 1 × 10^6^ of the other tumor cells used in this study.

In this study, most tumor cells cultured in glucose-containing media had specific rates of lactate production (qP Lactate) that were 15% to 100% higher under neutral conditions as compared to those under acidic conditions; likewise, qP Lactate was also 15% to 80% higher under hypoxic conditions as compared to normoxic conditions. Calu-1 had the highest qP Lactate when it was grown using glucose at a pH of 7.2 under normoxia (Glc/pH7.2/N); MRC-5 had the highest qP Lactate in comparison with the other tumor cell lines when it was grown in Glc/pH7.2/H, glucose-containing media at a pH of 6.2 and under normoxia (Glc/pH6.2/N), and glucose-containing media at a pH of 6.2 and under hypoxia (Glc/pH6.2/H) (Table 3 and Table 4). Interestingly, qP Lactate of MCR-5 cells was higher when they were grown in Glc/pH6.2/N as compared to when they were cultured in Glc/pH6.2/H, indicating that MRC-5 cells can produce lactate under normoxia as well as under hypoxia (Table 3). However, the lactate yield from glucose was higher in all the tumor cells compared to MRC-5 fibroblasts. The kinetic parameter qP Lactate is usually associated with qS Glucose because cells can produce more lactate when they consume more glucose.

### 3.3. Most Lung Cancer Cell Lines Consumed Lactate under Normoxia or Hypoxia

Because MCF-7 has been reported to consume lactate under normoxia [14], and it has not been previously reported whether lung cancer cells can consume lactate, we included MCF-7 as a positive control.

Except for Calu-1, we found that lung cancer cell lines and MRC-5 can consume lactate when cells are cultured in lactate-containing, glucose-free media. MRC-5, A427, SKMES-1, and MCF-7 consumed lactate during the exponential growth phase (Table 4, Figure 2 and Figure 3). Under normoxia, the lactate consumption of MRC-5, A427, and A549 was inhibited or diminished by acidosis (Table 3 and Table 4). However, most lung cancer cell lines could not proliferate when cultured in the Lac/pH7.2/H condition. A427, A549, MCF-7, and MRC-5 cells consumed lactate to survive under this condition.

### 3.4. Glutamine Consumption Diminished under Acidosis or Hypoxia

Although the dependence on glutamine during lung cancer cell line growth and survival has been reported [16], only A549 and A427 depleted glutamine (8 mM) after 72 h of culture under normoxia. This was independent of the presence of glucose or lactate. The other cell lines and the fibroblasts did not deplete glutamine. Our results agree with those of the study of Van den Heuvel et al. [17], finding that A549 and A247 have a higher dependence on glutamine for growth than SKMES-1. When culturing under glucose, pH 7.2, and hypoxia, MRC-5, A549, and SKMES-1 cells showed lower glutamine consumption than the normoxic condition. This phenomenon was also observed when most cell lines were cultured at pH 6.2.

When lactate was the primary carbon source, we found that the specific rates of glutamine consumption and glutamate production in the lung cancer cell lines and MCF-7 slightly increased compared with glucose-supplemented media (Table 3 and Table 4), but they were abolished when most cells were cultured under hypoxia.

Compared to neutral conditions, under acidosis, glutamine consumption and glutamate production diminished by 15% to 90% in most lung cancer cells (Figure 2 and Figure 3).

### 3.5. Lung Cancer Cells Increased CD98 Expression When Lactate Was the Predominant Carbon Source

During exponential growth, A549 and Calu-1 cells did not consume lactate in the absence of glucose. Instead, only a small amount of glutamine consumption was observed. Accordingly, we evaluated the expression of the amino acid transport CD98 by flow cytometry to determine if the reduced consumption of lactate and glutamine was related to amino acid consumption.

A427, Calu-1, and SKMES-1 cells significantly increased CD98 expression (as demonstrated by the increase in MFI) when lactate was the primary carbon source instead of glucose, irrespective of the pH or oxygen tension (Lac/pH7.2/N Lac/pH6.2/N, Lac/pH7.2/H and Lac/pH6.2/H conditions) (Figure 4A). Because most A427 and A549 cells expressed CD98, only Calu-1 cells increased the percentage of cells that expressed CD98 when lactate was present (Figure 4B). Furthermore, A549 cells significantly increased CD98 expression under hypoxia, independent of the carbon source (Figure 4A).

### 3.6. Acidosis Induced G0/G1 Arrest in A427 Cells

When we observed that lung tumor cells did not grow under the Lac/pH7.2/H condition, we analyzed the cell cycle of A427 cells to determine how the culture conditions affect the cell cycle profile and DNA synthesis.

Generally, compared to neutral conditions, acidosis increased the percentage of cells in the G0/G1 phase (Figure 5A, *p* < 0.05). Moreover, no change was observed in the cell death as measured by the subG0 peak. However, the combination of hypoxia and acidosis further increased the percentage of cells in the G0/G1 phase when glucose was the predominant carbon source (Figure 5A, *p* < 0.01). The change of glucose to lactate increased cell death (*p* < 0.001). In addition, when cells were grown under hypoxia, the synthesis phase was abrogated entirely (*p* < 0.001). Thus, acidosis induced an arrest in G0/G1 but, in combination with the hypoxia, the S-phase diminished, and cell death of A427 cells increased (Figure 5B). Accordingly, acidosis allowed the remaining cells to survive for more time, potentially allowing for survival until there is a change in the tumor microenvironment.

## 4. Discussion

In this study, we report how the specific growth rate, the specific consumption rates of glucose, lactate, and glutamine, and the specific rates of lactate and glutamate production were affected by changes in the microenvironment of lung tumor cells. Assessing for these kinetic parameters allowed for the determination of how much metabolite is consumed or produced per cell or 1 × 10^6^ cells. This normalization allowed us to compare different types of cells and growth conditions, thus providing a better understanding of cancer cell metabolism.

It has previously been reported that using a continuous-flow culture apparatus (Nutrostat) maintained a continuous proliferation of Jurkat cells. This study tested two different concentrations of glucose (10 mM and 0.75 mM), finding a doubling time of 24 h in media containing 10 mM glucose. Furthermore, the specific rate of glucose consumption (qS Glucose) was 0.23 µmol/10^6^ cells × h, and the specific rate of lactate production (qP Lactate) was 0.278 µmol/10^6^ cells × h [18]. We found that compared to Jurkat cells, lung cancer cell lines had a doubling time that ranged from 18 h to 30 h in media containing 10 mM glucose. The study by Birsoy et al. also documented slightly higher specific rates of glucose consumption and lactate production (qS Glucose (10 mM) = 0.38–0.84 µmol/10^6^ cells × h and qP Lactate = 0.77–2.68 μmol/10^6^ cells × h). This suggests that the lung cancer cell lines included in this study have a faster glucose metabolism than Jurkat cells. We also found that tumor cells’ specific growth rate (µ) was higher than the µ of the fibroblasts.

Furthermore, we found that MRC-5 had the highest specific rates of glucose consumption during the exponential phase of growth in comparison with both lung and breast tumor cell lines. This finding might appear contradictory, considering that most studies of non-transformed cells are based on non-proliferating cells in differentiated tissues. In vivo, normal cells carefully control the production and release of growth-promoting signals, ensuring cell growth and division according to tissue homeostasis [3]. FDG-PET is a helpful technique that measures tumor growth based on tumor cells’ high glucose consumption rates, increasing the signal in proportion to the tumor size. However, the specificity of this technique is only approximately 78%, which means that about 22% of the cases are false positives. This can occur when non-transformed cells have high glucose consumption rates, which can happen when cells are involved in inflammatory processes such as infections or injuries [19,20].

A metabolic symbiosis or parasitism relationship may also occur among cancer-associated fibroblasts (CAFs) and tumor cells. This relationship has been described and termed the “Reverse Warburg Effect”. In this phenomenon, CAFs, via aerobic glycolysis, provide the necessary fuels (l-lactate, ketones, amino acids, and fatty acids) to tumor cells [21,22]. On the other hand, our results indicate that aerobic glycolysis can take place in tumor cells and also in fibroblast cells, where MCR-5 cells also had a high specific rate of lactate production under both hypoxia and normoxia when they were cultured with glucose-containing media. Notably, lactate yield from glucose (YP/S) was higher in the tumor cells than in fibroblast cells, suggesting that tumor cells produce lactate from glucose.

Some authors have reported some kinetic characterization in tumor cells. For example, Wu et al. reported the proliferation curve, glucose consumption, and lactate production of a murine breast cancer cell (4T1) using 3 mM glucose with or without lactic acidosis under normoxia [23]. This study, however, did not calculate the specific growth rate or the consumption rates for these metabolites, nor did it analyze other conditions such as hypoxia or lactate with neutral pH. Kennedy et al. showed that MCF-7 cells consume lactate using an initial concentration of 20 mM but did not show the growth curve [14]. Xie et al. presented growth curves, glucose consumption, and lactate production for several tumor cell lines (4T1, Bcap37, HeLa, and A549) grown under two different culture conditions, one using 18 mM glucose and the other using glucose and lactic acidosis (20 mM/pH 6.7) under normoxia [24]. This study did not, however, report the kinetic parameters of these processes, nor did it analyze other conditions such as hypoxia or neutral pH. Liu et al. determined the effect of amino acid restriction on the rates of glucose consumption and lactate production by DU145 and PC3 prostate cancer cells using Met-, Tyr/Phe-, or Gln-medium [25]. This study determined the glucose consumption and lactate production rates every 24 h over four days without considering the growth phase of the culture. Also, Liu et al. did not report growth rates or include glucose consumption and lactate production rates for the Met+, Tyr/Phe+, and Gln+ medium.

Our results indicate that in the presence of glucose and hypoxia, the µ of lung cancer cells MCF-7 and MRC-5 slightly increased with acidosis in comparison to neutral conditions (except A549), while the specific rate of glucose consumption and lactate production diminished with acidosis in contrast with neutral conditions. Additionally, when the pH was neutral, MRC-5, A549, SKMES-1, and MCF-7 cells did not proliferate when cultured with lactate and under hypoxia. Instead, the cells remained viable for 48 h but began dying. However, when the pH was acidic, each cell line could proliferate. These results agree with our previous study [13]. Accordingly, acidosis potentially turns on a survival switch that might be a critical control point and amenable to cancer therapy.

Although Xie et al. reported that lactic acidosis considerably reduced the glucose consumption rate and virtually blocked net lactate generation in murine breast cancer cells (4T1), leading to a postulation that lactic acidosis could provide feedback to glucose utilization [24], our studies revealed that glucose and lactate consumption rates diminished due to the acidosis, independent of the levels of glucose or lactate. Kolesnik et al. evaluated the effect on proliferation and survival of lactic acidosis (pH 6.7) on only one lung cancer cell line from a mouse model (Lewis Lung Carcinoma Cells, LLC). They found that these conditions promoted the proliferation of LLC cells. The authors suggest that proliferation is supported by consuming glutamine [26]. Accordingly, our data show that lung cancer cells did not block lactate generation even under lactic acidosis; thus, lactate production is supported by amino acid consumption.

The regulation of ASCT2 and LAT1/2 expression is controlled by AMPK and mTOR, which are fundamental for energetic metabolism in cancer [11]. On the other hand, the regulation of CD98 expression has been mainly associated with cell spreading, migration, and proliferation [27]. Here, we show that the increased CD98 expression in lung cancer cell lines might be related to the absence of glucose in culture media independent of pH or oxygen tension. This conclusion is supported because the specific rate of glutamine consumption in lung cancer and MCF-7 cell lines was slightly increased in glucose-free media compared with glucose-supplemented media.

Several reports have indicated a rapid and uncontrolled proliferation of tumor cells. Traditional chemotherapeutic agents kill cells that divide rapidly [15,28]. Still, this kind of therapy has had limited success, especially in epithelial tumors of the breast, colon, and lung [28]. We found evidence that the growth rate of tumor cells is affected by the tumor microenvironment. When cells were cultured under hypoxia, their µ diminished compared to those cultured under normoxia. However, in response to acidosis, tumor cells turned on survival mechanisms that favor survival over proliferation. Wojtkowiak et al. reported that tumor cells cultured under acidosis increased cytoplasmic vacuolization, a process that was related to autophagy as a survival adaptation under acidosis [29]; using flow cytometry, we also found that cells cultured under acidosis had an increase in granularity.

Wu et al. previously reported that lactic acidosis promoted a G1/G0 cell-cycle arrest, autophagy induction, and apoptosis inhibition under glucose deprivation and normoxia conditions [23]. However, we found that the percentage of cells in G0/G1 increased by 5% to 20% when cultured under acidosis, even in a medium with glucose. It has been suggested that cancer cells found inside the hypoxic core of the tumor are frequently the cause of treatment failure, likely due to issues with drug inaccessibility [9]. Thus, we hypothesize that when acidosis develops in the tumor microenvironment, tumor cells induce a G0/G1 cell-cycle arrest, failing treatments that rely on cell proliferation. Furthermore, drugs that block glucose, lactate, and glutamine consumption will continue to have limited success, given that under acidosis, the specific rates of glucose, lactate, and glutamine consumption diminish. Consequently, we propose that targeting cells in the G0/G1 cell-cycle phase or targeting the acidosis present in the tumor microenvironment would be beneficial as a treatment for cancer.

## 5. Conclusions

This study elucidated adaptive cellular responses in tumor cells that occur due to changes in the microenvironment and result in a proliferative or survival advantage. We report that acidosis, especially in conjunction with hypoxia, turns on essential survival mechanisms, and consequentially, proliferation becomes a secondary objective. Understanding metabolic adaptive mechanisms for tumor cell survival may provide promising opportunities to improve traditional cancer therapies, considering that other conditions (lactic acidosis, hypoxia) should be tested when analyzing drug sensitivity.

## Figures and Tables

**Figure 1 metabolites-14-00103-f001:**
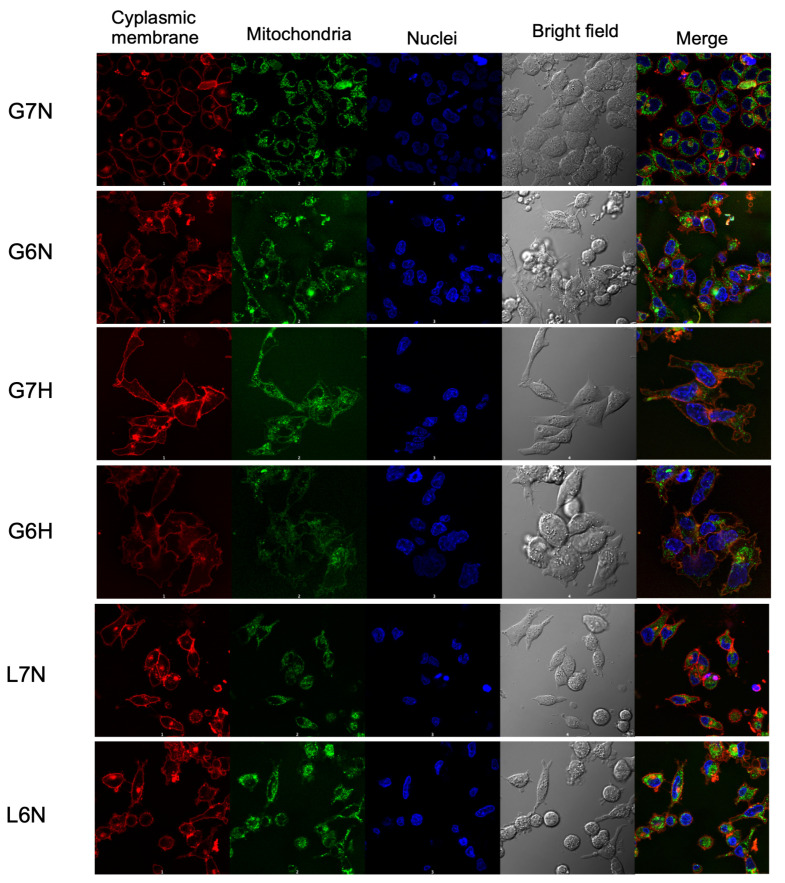
Representative confocal images (60×) of A-427 cells cultured using RPMI-1640 supplemented with glucose (10 mM) and at pH 7.2 or 6.2 under normoxia or hypoxia or using RPMI-1640 supplemented with lactate (20 mM) and at pH 7.2 or 6.2. Cells were stained with CellMask Orange (cytoplasmic membrane), MitoTracker Green (mitochondria), and Hoechst 33342 (nuclei) as in [13].

**Figure 2 metabolites-14-00103-f002:**
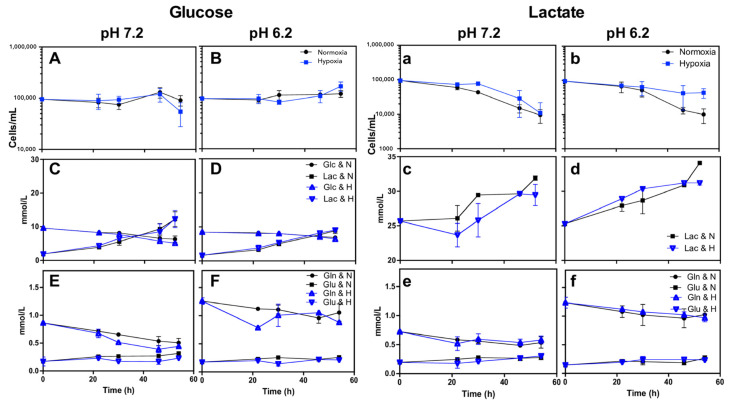
Characterization of Calu-1 using RPMI-1640 supplemented with glucose (10 mM) and at pH 7.2 (**A**,**C**,**E**) or pH 6.2 (**B**,**D**,**F**) or using RPMI-1640 supplemented with lactate (20 mM) and at pH 7.2 (**a**,**c**,**e**) or pH 6.2 (**b**,**d**,**f**). (**A**,**B**,**a**,**b**) Growth curve, (**C**,**D**) glucose consumption and lactate production, (**c**,**d**) lactate consumption or production, and (**E**,**F**,**e**,**f**) glutamine consumption and glutamate production under normoxia (21% O_2_) or hypoxia (2% O_2_). GIu, glucose; Lac, lactate; Gln, glutamine; Glu, glutamate.

**Figure 3 metabolites-14-00103-f003:**
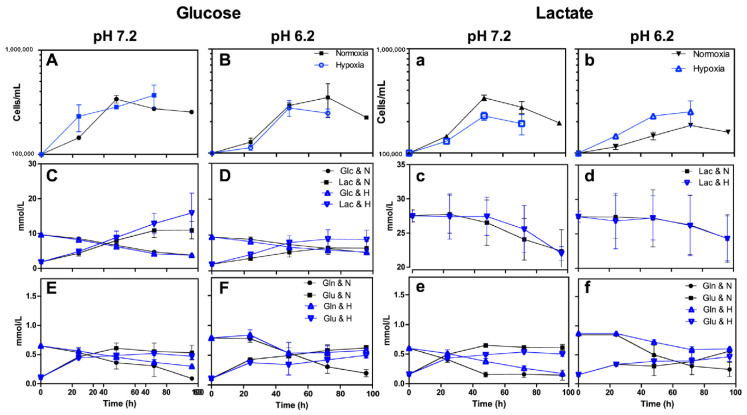
Characterization of SKMES-1 using RPMI-1640 supplemented with glucose (10 mM) and at pH 7.2 (**A**,**C**,**E**) or pH 6.2 (**B**,**D**,**F**) or using RPMI-1640 supplemented with lactate (20 mM) and at pH 7.2 (**a**,**c**,**e**) or pH 6.2 (**b**,**d**,**f**). (**A**,**B**,**a**,**b**) Growth curve, (**C**,**D**) glucose consumption and lactate production, (**c**,**d**) lactate consumption or production, and (**E**,**F**,**e**,**f**) glutamine consumption and glutamate production under normoxia (21% O_2_) or hypoxia (2% O_2_). Glc, glucose; Lac, lactate; Gln, glutamine; Glu, glutamate.

**Figure 4 metabolites-14-00103-f004:**
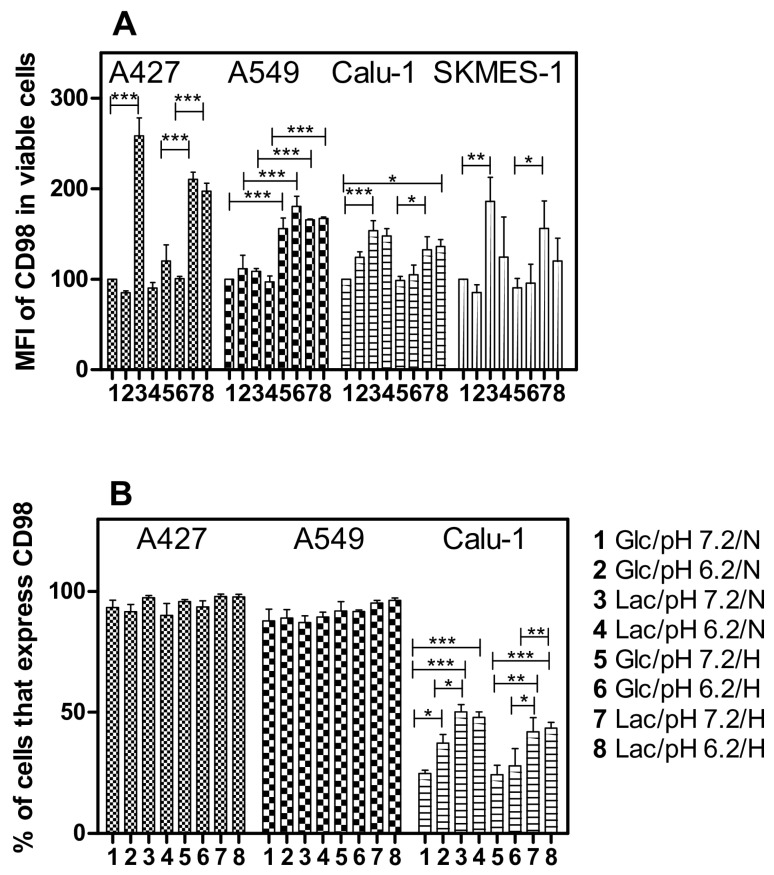
Median fluorescence intensity (MFI) of CD98 signal in lung cancer cell lines (**A**) and percentage of cells that express CD98 (**B**). Cells were grown under different culture conditions. Glc, glucose; Lac, lactate, N, normoxia, H, hypoxia; * *p* < 0.05; ** *p* < 0.01; *** *p* < 0.001.

**Figure 5 metabolites-14-00103-f005:**
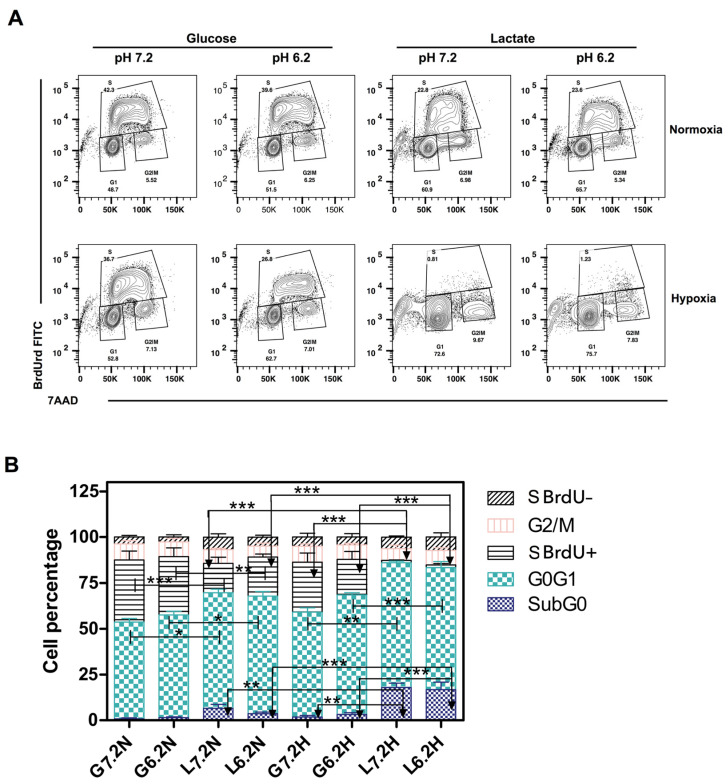
Cell cycle analysis of the adenocarcinoma cell line A-427 grown under eight different culture conditions. 5′-Bromo-2′deoxyuridine (BrdU, 10 μM) was added during the last hour of incubation. (**A**) BrdU vs. 7-amino-actinomycin D (7-AAD) contour-plot showing cell cycle phases and subG0 cells; and (**B**) the percentage of cells in the different phases of the cell cycle. * *p* < 0.05; ** *p* < 0.01; *** *p* < 0.001.

**Table 1 metabolites-14-00103-t001:** Growth parameters from different tumor cell lines and fibroblasts grown in a tissue-culture plate using RPMI-1640 supplemented with glucose (10 mM) and at pH 7.2 or 6.2 under normoxia or hypoxia.

		Glucose pH 7.2			Glucose pH 6.2		
Tumor Cell Line	%O_2_	Growth Phase Length (h)	Doubling Time (h)	μ (×10^−2^ h^−1^)	Growth Phase Length (h)	Doubling Time (h)	μ (×10^−2^ h^−1^)
MRC-5	N	24–96	46.11	1.5	24–72	105.2	0.66
	H	24–72	86.28	0.8	48–72	437.4	0.16
A427	N	0–72	30.46	2.28	0–48	19.44	3.57
	H	0–48	57.13	1.21	0–32	34.02	2.04
A549	N	0–45	18.75	3.7	6–69	29.71	2.33
	H	0–45	23.32	2.97	22–69	31.24	2.22
Calu-1	N	22–46	29.42	2.36	22–54	103.3	0.67
	H	22–46	54.56	1.27	22–54	33.22	2.09
SKMES-1	N	0–48	23.34	2.97	0–48	26.41	2.62
	H	0–72	50.1	1.38	0–48	26.18	2.65
MCF-7	N	0–48	31.11	2.23	0–48	40.69	1.7
	H	0–48	36.18	1.91	0–48	29.6	2.3

μ = Specific rate of growth determined under the exponential phase; N, normoxia = 21% O_2_; H, hypoxia = 2.0% O_2_.

**Table 2 metabolites-14-00103-t002:** Growth parameters from different tumor cell lines and fibroblast cells cultured using glucose-free RPMI-1640 supplemented with lactate (20 mM) and at pH 7.2 or 6.2 under normoxia or hypoxia.

		Lactate pH 7.2			Lactate pH 6.2		
Tumor Cell Line	%O_2_	Growth Phase Length (h)	Doubling Time (h)	μ (×10^−2^ h^−1^)	Growth Phase Length (h)	Doubling Time (h)	μ (×10^−2^ h^−1^)
MRC-5	N	24–96	36.4	1.9	24–72	61.78	1.12
	H	-	-	-	24–96	102.9	0.67
A427	N	0–32	32.44	2.14	0–48	31.95	2.17
	H	-	-	-	-	-	-
A549	N	6–69	35.87	1.9	22–69	22.86	3.03
	H	-	-	-	6–45	87.11	0.8
Calu-1	N	-	-	-	-	-	-
	H	-	-	-	-	-	-
SKMES-1	N	0–48	23.38	2.96	0–72	77.55	0.89
	H	0–48	36.95	1.88	0–48	39.66	1.75
MCF-7	N	0–48	31.84	2.18	0–72	55.65	1.25
	H	-	-	-	0–48	128.2	0.54

μ = Specific rate of growth determined under the exponential phase; N, normoxia = 21% O_2_; H, hypoxia = 2% O_2,_ - = non-determined because cells did not proliferate.

**Table 3 metabolites-14-00103-t003:** Kinetic parameters from different tumor cell lines and fibroblast cells, grown using RPMI-1640 supplemented with glucose (10 mM) and at pH 7.2 pH 6.2 under normoxia or hypoxia.

		Lactate pH 7.2				Lactate pH 6.2			
Tumor Cell Line	%O_2_	^a^ q_S Lactate_ (μmol/10^6^ cells × h)	^b^ q_P Lactate_ (μmol/10^6^ cells × h)	^c^ q_S Glutamine_ (μmol/10^6^ cells × h)	^d^ q_P Glutamate_ (μmol/10^6^ cells × h)	q_S Lactate_ (μmol/10^6^ cells × h)	q_P Lactate_ (μmol/10^6^ cells × h)	q_S Glutamine_ (μmol/10^6^ cells × h)	q_P Glutamate_ (μmol/10^6^ cells × h)
MRC-5	N	0.67	-	0.08	0	-	0.19	0.14	0.07
	H	-	-	-	-	1.25	-	0.01	0
A427	N	0.28	-	0.08	0.02	-	0.003	0.05	0.01
	H	-	-	-	-	-	-	-	-
A549	N	-	0.13	0.06	0.03	-	0.32	0.05	0.02
	H	-	-	-	-	-	0.24	0.03	0.02
Calu-1	N	-	-	-	-	-	-	-	-
	H	-	-	-	-	-	-	-	-
SKMES-1	N	0.13	-	0.05	0.06	0.13	-	0.04	0.04
	H	0.01	-	0.03	0.05	0.25	-	0	0.03
MCF-7	N	0.31	-	0.075	0.001	0.21	-	0.036	0.006
	H	-	-	-	-	-	0.095	0.014	0.017

(a) Specific rate of glucose consumption during the exponential phase. (b) Specific rate of lactate production during the exponential growth. (c) Specific rate of glutamine consumption during the exponential phase. (d) Specific rate of glutamate production during the exponential growth. N, normoxia = 21% O_2_. H, hypoxia = 2% O_2_. - = non-detected.

**Table 4 metabolites-14-00103-t004:** Specific glucose and glutamine consumption rates and lactate and glutamine production using lactate as the primary carbon source in cancer cell lines and fibroblasts.

Tumor Cell		Glucose pH 7.2				Glucose pH 6.2			
Line	%O_2_	^a^ q_S Glucose_	^b^ q_P Lactate_	^c^ q_S Glutamine_	^d^ q_P Glutamate_	q_S Glucose_	q_P Lactate_	q_S Glutamine_	q_P Glutamate_
		(μmol/10^6^ cells × h)	(μmol/10^6^ cells × h)	(μmol/10^6^ cells × h)	(μmol/10^6^ cells × h)	(μmol/10^6^ cells × h)	(μmol/10^6^ cells × h)	(μmol/10^6^ cells × h)	(μmol/10^6^ cells × h)
MRC-5	N	0.91	1.25	0.07	0	1.49	1.94	0.03	0.001
	H	1.28	1.96	0.02	0.08	1.11	1.7	0	0
A427	N	0.43	1.19	0.05	0.01	0.43	0.72	0.08	0.001
	H	0.87	1.91	0.07	0.02	0.65	1.32	0.06	0.014
A549	N	0.58	0.89	0.1	0.02	0.2	0.42	0.06	0.01
	H	0.52	1.04	0.01	0.02	0.09	0.64	0.06	0.015
Calu-1	N	0.84	2.68	0.08	0.005	0.32	1.33	0.01	0.007
	H	1.06	1.75	0.12	0	0.49	1.55	0	0.004
SKMES-1	N	0.38	0.77	0.036	0.06	0.3	0.47	0.03	0.054
	H	0.28	0.56	0.014	0.02	0.44	0.93	0.04	0.036
MCF-7	N	0.3	0.5	0.052	0	0.14	0.41	0.094	0
	H	0.46	0.91	0.051	0	0.34	0.78	0.062	0.003

(a) Specific rate of lactate consumption during the exponential phase. (b) Specific rate of lactate production during the exponential growth. (c) Specific rate of glutamine consumption during the exponential phase. (d) Specific rate of glutamate production during the exponential growth. N, normoxia = 21% O_2_. H, hypoxia = 2% O_2_.

## Data Availability

Data and materials are available upon reasonable request.

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
