# Peer review of "Metabolic Responses of Lung Adenocarcinoma Cells to Survive under Stressful Conditions Associated with Tumor Microenvironment"

_metabolites, 2024, doi:10.3390/metabo14020103_

Round 1

Reviewer 1 Report

Comments and Suggestions for Authors

This paper was provided reliable metabolites analyzing in cancer cells under microenvironment changes. The also provide hint(s) about the cancer cell targeting for limiting of nutrition may not a good method. There are several points needed to revise as following:    

1. Line 65 “also? did not report growth rates or include glucose consumption” should be delete “?”.

2. Line 242 : subtitle “Specific rates of glucose consumption and lactate production of tumor cells and fibroblast diminished under acidosis.” But MRC-5 fibroblast cannot be found big changes of glucose consumption and lactate production between pH 7.2 and pH 6.2 in Table. Therefore, subtopic should be corrected as “Specific rates of glucose consumption and lactate production of tumor cells diminished under acidosis”

3. Line 295 : subtitle “Glutamine consumption diminished under hypoxia or acidosis.” But in line 296-306 didn’t mention any sentence about the glutamine consumption under hypoxia condition.

Author Response

We are grateful for the reviewer's comments, and we have included our point-by-point response.

Reviewer 1

This paper provided reliable metabolite analysis in cancer cells under microenvironment changes. They also provide hints about the cancer cell targeting for limiting nutrition may not be a good method. There are several points that need to revise as follows:     

  1. Line 65 “also? did not report growth rates or include glucose consumption” should be deleted “?”.

The error was corrected. Thank you for your observation.

2. Line 242: subtitle “Specific rates of glucose consumption and lactate production of tumor cells and fibroblast diminished under acidosis.” But MRC-5 fibroblast cannot be found big changes of glucose consumption and lactate production between pH 7.2 and pH 6.2 in Table. Therefore, subtopic should be corrected as “Specific rates of glucose consumption and lactate production of tumor cells diminished under acidosis” 

This correction was also made and highlighted.

3. Line 295: subtitle “Glutamine consumption diminished under hypoxia or acidosis.” But in lines 296-306 didn’t mention any sentence about glutamine consumption under hypoxia conditions.

We apologize for this mistake the information was included in the corresponding section, which is highlighted.

Reviewer 2 Report

Comments and Suggestions for Authors

1)      The word “solid tumor” in the abstract could confuse the audience. The authors used 2D culture (monolayer) of lung cancer cell lines, not spheroid nor isolated tumors.

2)      I strongly advise adding a graphical abstract at the end of the introduction.

3)      I see the authors discussed their results, and compared them with previous research on the introductory parts (lines 49-66). I believe this part (lines 49-66) should be moved to the discussion.

4)      Why the authors used MCF-7 and MRC-5 cell lines? (the reason must be mentioned manuscript since the title of the paper is lung adenocarcinoma)

5)      The authors should mention in the manuscript (for section 2.5) how they for samples to determine metabolites from culture media. From the cells? And frequency of sampling (from time zero) should be mentioned.

6)      Why do the authors use CD 98 expression on lung cancer cell lines? Usually, CD 151 is expressed in 98% of non-small cancer cell lines.

7)      The quality of Figure 2 is not good at all.

8)      The authors should determine what is a stressful condition. Why study of lung cancer cells in stressful conditions is important since in nature, the tumor cells are not in stressful conditions unless they have been treated with anti-cancer agents.

9)      It would be nice if the authors provide images by confocal microscopy from cancer cells under different conditions of investigated glucose, lactate, hypoxia, normoxia, etc. to enrich your results. 

Author Response

We are grateful for the reviewer's comments, and we have included our point-by-point response.

1)      The word “solid tumor” in the abstract could confuse the audience. The authors used 2D culture (monolayer) of lung cancer cell lines, not spheroid nor isolated tumors.

Thank you for the observation. We have made a precision, indicating that tumor cell lines were cultured in monolayers

2)      I strongly advise adding a graphical abstract at the end of the introduction.

As suggested, we have added a graphical abstract. 

3)      I see the authors discussed their results, and compared them with previous research on the introductory parts (lines 49-66). I believe this part (lines 49-66) should be moved to the discussion.

As suggested by the reviewer, we incorporated the above-mentioned part into the Discussion section. 

4)      Why the authors used MCF-7 and MRC-5 cell lines? (the reason must be mentioned in the manuscript since the title of the paper is lung adenocarcinoma)

MCF-7 cell line is a breast cancer cell line, that was included as a control because it consumes lactate in the absence of glucose. MRC-5 fibroblasts are proliferative, non-transformed cells; for that reason, they were included as a control. This information has been included in the manuscript.

5)      The authors should mention in the manuscript (for section 2.5) how they sampled to determine metabolites from culture media. From the cells? And frequency of sampling (from time zero) should be mentioned.

The supernatant from one well of the 24-well plates was removed every 8, 12, or 24 hours, depending on the cell line, until 96 hours were completed. After measuring the volume, cell-free supernatants were stored at −20 ºC for subsequent analysis. A sample of the initial culture medium was stored at −20 º C and considered time zero in the metabolites’ analysis. This part was incorporated into the text.

6)      Why do the authors use CD 98 expression on lung cancer cell lines? Usually, CD 151 is expressed in 98% of non-small cancer cell lines.

CD151 is a molecule expressed in the lung that interacts with laminin-binding integrins and some growth factor receptors. Although this molecule is of particular interest for studying metastasis and angiogenesis, our study focused on the involvement of the metabolism of tumor cells. Hence, we focused on the expression of the amino acid transporter CD98.

7)      The quality of Figure 2 is not good at all.

We replaced Figure 2 with a higher-quality version. We apologize for this inconvenience.

8)      The authors should determine what is a stressful condition. Why study of lung cancer cells in stressful conditions is important since in nature, the tumor cells are not in stressful conditions unless they have been treated with anti-cancer agents.

The conditions we use to treat the tumor cells can be stressful because these are not usually found in homeostatic conditions. High lactate levels, for instance, alter the metabolism of the cells; even among tumor cells, they alter the metabolism and must adapt to survive. A low pH can be deleterious for stromal cells and alter cell proliferation, as shown in Fig 5. Environmental pressure due to nutrient deprivation, and hypoxia are environmental pressures that favor the selection of tumor clones that adapt to these conditions (http://dx.doi.org/10.1016/j.bbabio.2017.02.006). 

 9)      It would be nice if the authors provide images by confocal microscopy from cancer cells under different conditions of investigated glucose, lactate, hypoxia, normoxia, etc. to enrich your results. 

We incorporated a new figure (Fig. 1) showing morphological changes in A-427 cells cultured under different conditions. 

Reviewer 3 Report

Comments and Suggestions for Authors

The authors present an analysis of metabolic responses of lung adenocarcinoma cells under stressful conditions to identify changes associated with tumor microenvironment. The paper is overall well-written and some minor comments are presented below. 

FIGURES

In Figure 3 the x axes do not have labels to indicate the meaning of 1-8

INTRODUCTION

Hypoxia and its relationship with the tumor microenvironment is a well-described area of research.  The references for this manuscript are rather limited, the authors could consider expanding the cited literature.

https://doi.org/10.3390/cancers12113244

https://doi.org/10.3389/fimmu.2016.00052

https://doi.org/10.1016/j.ebiom.2021.103627

METHODS

Is the methodology to calculate the kinetics novel? If so this should be stated more clearly through the paper. If it is not, citations should be given. My assumption is that these are well-described techniques.

DISCUSSION

Same comment as above, acidosis is well-described and understood. Whilst the authors have measured kinetics, how much additional significance can be placed on the findings?

https://doi.org/10.1146/annurev-physiol-021119-034627

https://doi.org/10.1158/0008-5472.CAN-12-2796

Especially this sentence:

"Understanding metabolic adaptive mechanisms for tumor cell survival may provide promising opportunities to improve traditional cancer therapies."

Work has certainly been done in this area. Why has it met with so little success so far? Will this work enhance research efforts by providing a methodology to quantify kinetics? How does this compare with other TME work, for example TMEs for other cancer cell lines?

Overall, the paper is well written and the science appears to be sound. The findings, however, seem well-described and well-understood, even to a non-expert in the field. If this is genuinely the first time that anyone has done cell kinetics work under hypoxia for any cell line, I think this is very surprising. If the novelty is that no-one has done it for this specific case (lung cancer cell line, with hypoxia and acidosis varied) then the Discussion needs more comparison with what has been seen in other TMEs.

Comments on the Quality of English Language

English is very well written.

Author Response

The authors present an analysis of metabolic responses of lung adenocarcinoma cells under stressful conditions to identify changes associated with tumor microenvironment. The paper is overall well-written and some minor comments are presented below. 

We are in debt with the reviewer’s kind comments and suggestions.

FIGURES

In Figure 3 the x-axes do not have labels to indicate the meaning of 1-8

The labels indicating the meaning of 1-8 are located at the right of the bars.

INTRODUCTION

Hypoxia and its relationship with the tumor microenvironment is a well-described area of research.  The references for this manuscript are rather limited, the authors could consider expanding the cited literature.

https://doi.org/10.3390/cancers12113244

https://doi.org/10.3389/fimmu.2016.00052

https://doi.org/10.1016/j.ebiom.2021.103627

As suggested, we updated some of the references regarding the role of hypoxia and lactic acidosis. 

METHODS

Is the methodology to calculate the kinetics novel? If so this should be stated more clearly through the paper. If it is not, citations should be given. My assumption is that these are well-described techniques.

The reference as to how we calculated the kinetic parameters was included in the Methods section. 

DISCUSSION

Same comment as above, acidosis is well-described and understood. Whilst the authors have measured kinetics, how much additional significance can be placed on the findings?

https://doi.org/10.1146/annurev-physiol-021119-034627

https://doi.org/10.1158/0008-5472.CAN-12-2796

Especially this sentence:

"Understanding metabolic adaptive mechanisms for tumor cell survival may provide promising opportunities to improve traditional cancer therapies."

Work has certainly been done in this area. Why has it met with so little success so far? Will this work enhance research efforts by providing a methodology to quantify kinetics? How does this compare with other TME work, for example TMEs for other cancer cell lines?

Overall, the paper is well written and the science appears to be sound. The findings, however, seem well-described and well-understood, even to a non-expert in the field. If this is genuinely the first time that anyone has done cell kinetics work under hypoxia for any cell line, I think this is very surprising. If the novelty is that no-one has done it for this specific case (lung cancer cell line, with hypoxia and acidosis varied) then the Discussion needs more comparison with what has been seen in other TMEs.

Concerning the reviewer’s concerns, and contrary to what one might think, the characterization of the proliferation rate and consumption and production rates has been scarcely studied. In the Discussion section, we highlight the sections where other studies compare with conditions similar to ours. We delimited the conclusion by proposing that lactic acidosis and hypoxia are TME conditions that should be considered when evaluating drug sensitivity in vitro. 

Round 2

Reviewer 2 Report

Comments and Suggestions for Authors

The authors address my questions and I recommend publishing